# Integration of a physiologically-based pharmacokinetic model with a whole-body, organ-resolved genome-scale model for characterization of ethanol and acetaldehyde metabolism

Leo Zhu[1], William Pei[1,2], Ines Thiele[3,4,5]*, Radhakrishnan Mahadevan[1,2]*

**1** Department of Chemical Engineering and Applied Chemistry, University of Toronto, Toronto, Ontario, Canada, **2** Institute of Biomedical Engineering, University of Toronto, Toronto, Ontario, Canada, **3** School of Medicine, National University of Ireland at Galway, Galway, Ireland, **4** Discipline of Microbiology, National University of Ireland at Galway, Galway, Ireland, **5** APC Microbiome Ireland, Cork, Ireland

* ines.thiele@nuigalway.ie (IT); krishna.mahadevan@utoronto.ca (RM)

**Data Availability Statement:** Code is available on Github https://github.com/LMSE/HH-PBPK-Ethanol.

## Abstract

Ethanol is one of the most widely used recreational substances in the world and due to its ubiquitous use, ethanol abuse has been the cause of over 3.3 million deaths each year. In addition to its effects, ethanol's primary metabolite, acetaldehyde, is a carcinogen that can cause symptoms of facial flushing, headaches, and nausea. How strongly ethanol or acetaldehyde affects an individual depends highly on the genetic polymorphisms of certain genes. In particular, the genetic polymorphisms of mitochondrial aldehyde dehydrogenase, ALDH2, play a large role in the metabolism of acetaldehyde. Thus, it is important to characterize how genetic variations can lead to different exposures and responses to ethanol and acetaldehyde. While the pharmacokinetics of ethanol metabolism through alcohol dehydrogenase have been thoroughly explored in previous studies, in this paper, we combined a base physiologically-based pharmacokinetic (PBPK) model with a whole-body genome-scale model (WBM) to gain further insight into the effect of other less explored processes and genetic variations on ethanol metabolism. This combined model was fit to clinical data and used to show the effect of alcohol concentrations, organ damage, ALDH2 enzyme polymorphisms, and ALDH2-inhibiting drug disulfiram on ethanol and acetaldehyde exposure. Through estimating the reaction rates of auxiliary processes with dynamic Flux Balance Analysis, The PBPK-WBM was able to navigate around a lack of kinetic constants traditionally associated with PK modelling and demonstrate the compensatory effects of the body in response to decreased liver enzyme expression. Additionally, the model demonstrated that acetaldehyde exposure increased with higher dosages of disulfiram and decreased ALDH2 efficiency, and that moderate consumption rates of ethanol could lead to unexpected accumulations in acetaldehyde. This modelling framework combines the comprehensive steady-state analyses from genome-scale models with the dynamics of traditional PK models to create a highly personalized form of PBPK modelling that can push the boundaries of precision medicine.

**Funding:** Authors acknowledge funding from Natural Sciences and Engineering Research Council of Canada through the Discovery Grant program (to RM) and the M3 CREATE program (to LZ) and from the European Research Council (ERC) under the European Union's Horizon 2020 research and innovation programme (grant agreement No 757922) (to IT). The funders had no role in study design, data collection and analysis, decision to publish, or preparation of the manuscript.

**Competing interests:** The authors have declared that no competing interests exist.

## Author summary

Alcohol is a widely used recreational drug in many parts of the world and it is often abused or misused, leading to the deaths of millions of people each year from driving under the influence and overdose. Additionally, the body breaks down alcohol into acetaldehyde, a carcinogen that has its own effects ranging from headaches and nausea to liver damage. The effects of ethanol and acetaldehyde vary due to genetic variations that create different forms of the enzymes responsible for breaking them down. Due to these differences, it is important to characterize how these changes affect the metabolism of alcohol and acetaldehyde. To capture these differences, we have created a new model that integrates the traditional pharmacokinetic model with a whole-body genome-scale model that can characterize different genetic variations. In addition, traditional models often require experimentally measured data, yet with this new framework we avoid this tedious process by mathematically solving the genome-scale model with the dynamic Flux Balance Analysis technique, allowing for gap filling. Through this model, we show that the whole-body genome-scale model demonstrates flexibility and robustness that has not been seen before in pharmacokinetic models. Our model combines advantages from both pharmacokinetic and genome-scale modelling and can be personalized to characterize individual reactions to other drugs and further precision medicine.

## Introduction

Ethanol is a drug that has been extensively studied and is widely used in the world today. Ethanol abuse can lead to dependence, liver cirrhosis, social withdrawal, and serious implications when driving under the influence, leading to approximately 3.3 million deaths each year [1]. This alarming number has encouraged researchers to develop mathematical models to better understand ethanol metabolism [2]. It is understood that ethanol is primarily metabolized by liver alcohol dehydrogenase (ADH) into acetaldehyde, which is in turn eliminated by mitochondrial aldehyde dehydrogenase (ALDH2) into acetate [3,4]. This is the primary metabolism pathway included in most ethanol metabolism models, however there are other processes that are frequently omitted in modelling, including microsomal ethanol oxidizing systems (namely cytochrome P450 enzyme 2E1), catalase, peroxisomes, and non-oxidative methods such as conversion by fatty acid ethyl esters [3–6].

Acetaldehyde, the primary metabolite of ethanol, is toxic. Its build-up can cause facial flushing, headaches, nausea, dizziness, and tachycardia [7]. The efficacy of its elimination by ALDH2 is heavily influenced by genetic polymorphisms, and notably East Asian populations with the ALDH2*2 genotypes have almost no ALDH2 activity when compared to the wild type, allowing acetaldehyde to accumulate and cause "flushing" symptoms to occur [8–11]. Interestingly, Disulfiram (Antabuse), and other drugs aimed at reducing alcohol dependence, purposefully inhibit ALDH2 to produce the same symptoms and cause oversensitivity to ethanol [12–15]. Acetaldehyde exposure has been implicated in increased risk for a variety of cancers [16–19] and may also be involved in alcohol hangovers [20–23]. Thus, it is important to characterize how different populations' genetic variations and drinking habits can lead to various exposure and responses to ethanol and acetaldehyde.

The pharmacokinetics of ethanol through alcohol dehydrogenase have been well characterized in literature since Widmark's research in 1933 assuming 0th order elimination [24]. This work was expanded upon by Lundquist and Wolthers, who incorporated Michaelis-Menten

Kinetics to describe the non-linearity of elimination curves below 0.2mg/mL [25]. Subsequently, further research on ethanol metabolism investigated the distribution of ethanol in the blood proportional to total body water, the effect of first-pass metabolism by the stomach, and extrahepatic pathways for ethanol clearance [26–31]. This work continues today in the form of physiologically-based pharmacokinetic (PBPK) models, one of which we have previously developed [32]. We used this model to capture the absorption, distribution, metabolism, and elimination (ADME) of ethanol, different metabolic mechanisms, aging effects, biochemical variation in enzyme activity, sex differences, and the "meal effect" on ethanol metabolism [32].

PBPK models are useful when enzyme kinetics and transport rates are known, however in the absence of such data researchers often resort to fitting various parameters to clinical data, leading to overfitting and model stiffness. However, genome-scale models (GEM) can remedy this gap by predicting the flux of many reactions when subject to certain constraints [33]. In this paper, we augment the previously developed PBPK model further by integrating a whole-body, organ-resolved, sex-specific genome-scale model (WBM) to provide further insight into traditionally ignored processes and how inter-individual differences can lead to variations in the metabolism of ethanol and acetaldehyde, ultimately navigating towards precision medicine.

Precision medicine seeks to create personalized computational models of the human body to predict the impact of various therapeutic approaches [33]. Using the constraint-based reconstruction and analysis approach (COBRA), the Harvey-Harvetta WBM developed by Thiele et al expands upon the molecular networks in previous human GEMs [34–35] by integrating organ anatomy and physiology [36–37]. Within the male Harvey reconstruction, the biochemical reactions governed by genetics are represented in a stoichiometric matrix (81094 reactions and 56452 metabolites), and the Flux Balance Analysis (FBA) technique is applied to solve for steady-state reaction fluxes given a specified objective function [38]. Here, we employ dynamic FBA [39] on the Harvey WBM to perform unsteady-state characterization of ethanol metabolism, pharmacokinetics, and pharmacodynamics. By combining the PBPK model, which predicts the kinetics of ethanol distribution throughout the body, with the WBM, which predicts steady-state ethanol metabolism through multiple pathways at the organ and molecular levels, we are able to harness the benefits of both model types to create a framework that allows for personalized predictions of ethanol metabolism, and potentially other metabolites of interest in the future.

## Methods

### Physiologically-Based Pharmacokinetic (PBPK) model development

The PBPK model used to track the distribution of ethanol concentrations expands on a previously published model [32] by having individual compartments for the stomach, small intestines, and large intestines instead of a single gut compartment. This separation distinguishes between those organs' tissue and lumen for better predictions of absorption kinetics [32]. This model breaks down the human body into 14 tissue compartments (consisting of the lung, liver, stomach, small intestines, large intestines, pancreas, spleen, kidney, skin, muscle, adipose, brain, heart, and blood) and 3 luminal compartments (consisting of the stomach, small intestines, and large intestines) (Fig 1).

Within the model, an individual's age, sex, height, weight, and body fat percentage are taken as model inputs to estimate the mass of various tissue compartments as well as the blood flow to each organ [40,41]. Based on the assumption that the partitioning of metabolites between tissue and plasma is in equilibrium, the metabolite lipophilicity and the unbounded metabolite fraction are used to calculate tissue-plasma partition coefficients based on tissue

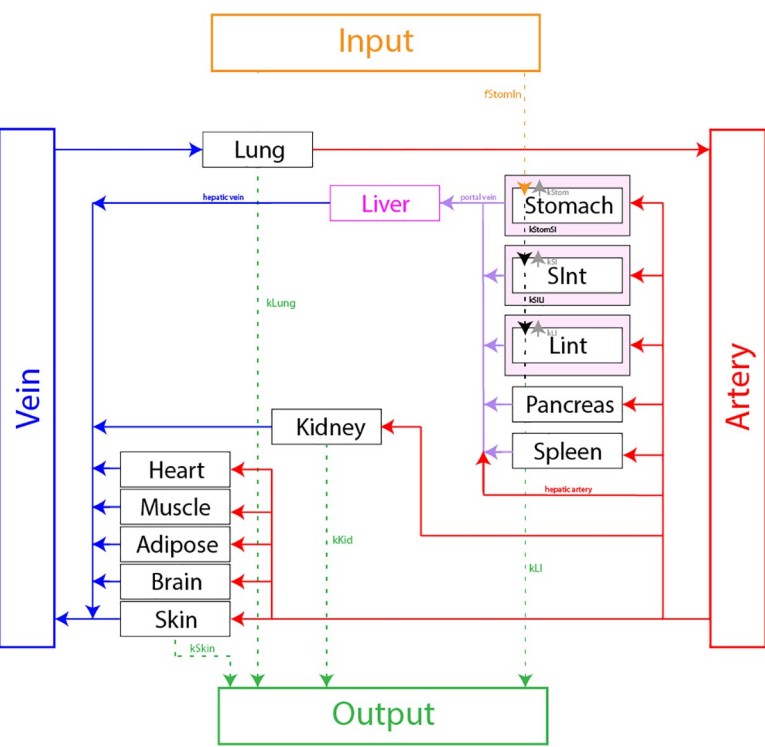

**Fig 1. Schematic of the PBPK model.** The red, blue, and purple lines denote arterial, venous, and hepatic portal blood flows, respectively. There are two mixing points in the model at the lung and at the liver to account for their respective physiology. The model includes an input at the stomach, and outputs at the skin, kidney, lung, and large intestines.

proportions of water, neutral lipids, and phospholipids [42,43]. For more information about the physical properties please visit Table 1.

The model was formulated as a system of 34 ordinary differential equations (ODEs), which represented both the transport of metabolites (ethanol, acetaldehyde) across the various compartments as well as enzyme activity via Michaelis-Menten kinetics. *Eq 2.1-2.3* show the development of the ODEs from mass balance. These ODEs characterize the concentration changes in each organ and were solved simultaneously using numerical integration on MATLAB (Version 9.6.0.1174912 R2019a) at each time step.

In these ODEs, organ blood flow ($Q_i$) and organ volume ($V_i$) were physiological parameters calculated by correlations with age, sex, height, weight, and body fat percentage [39]. The tissue partition coefficient ($K_i$) represents the pharmacokinetic properties of metabolites in the various compartments and was calculated based on the metabolite's lipophilicity, fraction unbound in the blood, and tissue composition [42]. The absorption rate constant ($k_i$) refers to the first-order absorption of metabolites from the organs and the transport rate constants ($k_{ij}$) refer to the transport of Ethanol between organs. $R_i$ refers to any reactions that may occur in the tissue, characterized by either Michaelis-Menten kinetics or the genome-scale model.

**Table 1. Pharmacokinetic parameters for ethanol and acetaldehyde.**

| Parameter | Ethanol | Acetaldehyde | Unit |
|---|---|---|---|
| Molecular Weight | 46.07 | 44.05 | g/mol |
| Lipophilicity | -0.31 | -0.34 | |
| Fraction Unbound | 0.99 | 0.99 | |

**Table 2. Parameters used for the model ODE development.**

| Parameter | Definition | Units | Source |
|---|---|---|---|
| $C_i$ | Concentration of metabolite in organ $i$ | mM | Experimental |
| $Q_i$ | Blood flow rate to organ $i$ | L/min | [40] |
| $V_i$ | Volume of organ $i$ | L | [40] |
| $K_i$ | Partition coefficient between organ $I$ and blood compartment | mM/mM | [42] |
| $k_i$ | Absorption rate of metabolite in organ $i$ | 1/min | Fitted |
| $k_{ij}$ | Transport rate of metabolite between organs $i$ and $j$ | 1/min | Fitted |
| $R_i$ | Metabolism of metabolite in organ $i$ | mM/min | [37] |
| $V_{max}$ | $V_{max}$ from Michaelis-Menten kinetics | mM/min | [11] |
| $K_m$ | Michaelis-Menten constant | mM | [11] |

These variables are tabulated in Table 2. The generic equation for the compartments is shown in *Eq 2.3*. For the complete set of tissue-specific equations, please refer to (S1 Text).

$$accumulation = in - out + generation - consumed \tag{1}$$

$$\Delta concentration = artery - vein - excretion - metabolized \tag{2}$$

$$\frac{dC_i}{dt} = \frac{Q_i}{V_i}\left(C_{blood} - \frac{C_i}{K_i}\right) - k_i C_i - R_i, \ \text{where} \ R_i = \left(\frac{V_{max}C}{K_m + C}\right)_i \tag{3}$$

## Whole-body metabolic model development

The male WBM model, Harvey, consists of 81094 reactions and 56452 metabolites [37]. Here, we focused on the exchange and metabolism of ethanol and acetaldehyde (Fig 2). To generate this schematic, all ethanol and acetaldehyde species were identified in the model, and related reactions were found with the function 'findRxnsFromMets' in COBRA Toolbox [36]. The full list of reactions can be found in (S1 and S2 Tables).

## PBPK-WBM integration

The PBPK model and WBM have slight differences in their metabolites inputs and outputs, thus additional reactions were added using the COBRA Toolbox function 'addReaction' for consistency. The list of ethanol-related reactions and their contributions to overall ethanol metabolism can be found in Table 3. While ethanol had many ways of elimination, all the acetaldehyde species were eliminated by ALDH2 in the colon and liver (see S2 Table).

Traditionally FBA only solves for steady state fluxes, however the results of FBA can be used in a dynamic model to better understand the kinetics of ethanol metabolism. Integration of the two models employed dynamic FBA (see Eq 2.4), in which an objective function is maximized subject to a set of constraints [38]. Here, the Michaelis-Menten parameters in the PBPK model were used to set boundaries for the WBM liver alcohol dehydrogenase (ADH) and aldehyde dehydrogenase (ALDH2) reactions [47]. The objective was to maximize the amount of ethanol removed by the liver alcohol dehydrogenase ('Liver_ALCD2if'), and given that the liver accounts for 90–95% of total ethanol metabolism, the boundaries for the other metabolic and excretion reactions were calculated and constrained based on the liver ethanol metabolism rate [44–47].

To solve the WBM model, FBA was performed with the function 'SolveCobraLPCPLEX', and the flux values from the FBA solution were used in a dynamic FBA fashion to update the PBPK model for the next time step [39]. The FBA flux solutions were continuously used in the solver until their values deviated from the Michaelis-Menten rate past an acceptable tolerance

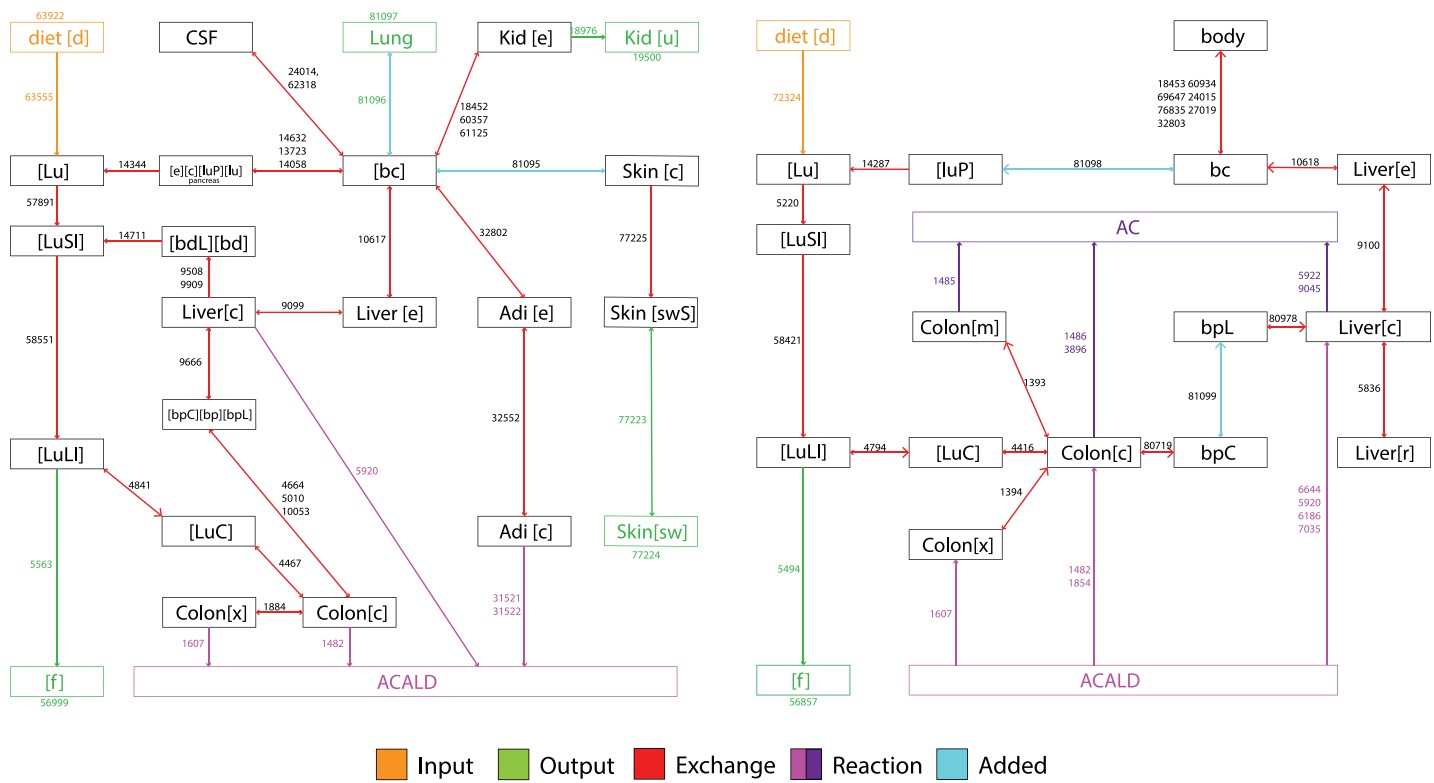

**Fig 2. WBM Schematics. (a)** Schematic for ethanol metabolism into acetaldehyde (ACALD) in the male WBM model, Harvey. **(b)** Schematic for acetaldehyde (ACALD) metabolism to acetate (AC) in Harvey. The orange lines denote model inputs and the green lines denote model outputs. The magenta and purple lines denote metabolism, and the blue lines are reactions that were added to Harvey to ensure consistency with the PBPK model. The numbers accompanying the arrows are the reaction number in the model. For a complete list of reactions please refer to the S1 and S2 Tables.

range (typically 1–2.5%), shown by Eq 2.5, after which the WBM was solved again to update the fluxes in the PBPK model. A schematic of this process can be found in S1 Fig.

$$max \, f^T v \, s.t. Sv = 0 \, lb_i < v_i < ub_i \, where \, ub_i = \frac{V_{max}C_i}{K_m + C_i} \tag{4}$$

$$\left| \frac{r_{MM} - r_{FBA}}{r_{MM}} \right| < tol \tag{5}$$

**Table 3. Reactions involved in ethanol metabolism and elimination in the whole-body genome-scale metabolic model (Harvey).** *Values were implemented as the lower and upper bounds of the corresponding reactions in the WBM.

| Harvey reaction abbreviation | PBPK model reaction name | % of total metabolism* | Notes |
|---|---|---|---|
| 'Diet_EX_etoh[d]' | Input | | Diet intake |
| 'EX_etoh_[br]' | kLungout | 0.05% | Lung/breath [31,44] |
| 'EX_etoh[u]' | kKid | 3–10% | Urine excretion [44] |
| 'EX_etoh[sw]' | kSkin | 3–10% | Sweat excretion [44] |
| 'Excretion_EX_etoh[fe]' | kLI | 0 | Feces excretion [44] |
| 'Liver_ALCD2if' | rliv | 90–95% | Liver alcohol dehydrogenase [27,44,45,46] |
| 'Colon_CAT2p' | rLI | 0–2% | Colon catalase [6] |
| 'Colon_ALCD2if' 'Adipocytes_ALCD2if' 'Adipocytes_ALCD2yf' | 0 | 0 | Colon/adipose alcohol dehydrogenase [46] |

## Results

### Predicting the effect of drink concentrations

As we had extended a previously published PBPK model [32] with more detailed gut components (i.e., stomach, small intestines, large intestines, and luminal compartments), we fitted this PBPK model to ethanol absorption data from Mitchell et al (n = 15) [47], thereby developing correlations between drink ethanol concentration (w/w), absorption rate, and transport rate through the gastrointestinal tract (Fig 3). Ethanol is primarily absorbed by the stomach and the small intestines, so we fitted the corresponding absorption rates, $k_{Stom}$ and $k_{SI}$, and the transport rate from the stomach to the small intestines, $k_{StomSI}$, to achieve this correlation [31,44] (S2 Text). The Mean Absolute Error (MAE) for the fitted beer (5.1%), wine (12.5%), and spirits (20%) simulations were 0.333, 0.6066, and 0.6858, respectively. Furthermore, the model's Areas Under the Curve ($AUC_{0-\infty}$) were 1612, 1861, and 2024 mM*min compared to Mitchell et al $AUC_{0-\infty}$ of 1502, 1697, and 1871 mM*min.

### Predicting acetaldehyde exposure

After fitting the PBPK model for ethanol absorption and clearance, we integrated the WBM to create the PBPK-WBM model and investigated the ability of the model to also predict acetaldehyde exposure. Umulis et al. created a PK model to predict both ethanol and acetaldehyde concentrations based on the results to a clinical study by Jones et al (n = 10). [11,48]. To ensure consistency, we used the Michaelis-Menten parameters taken from the Umulis et al. model and determined the elimination profile of ADH and ALDH2 (Fig 4). The model produced AUCs of 526.14 mM*min and 377.72 uM*min for ethanol and acetaldehyde, respectively, while the experimental data from Jones et al. had AUCs of 538.5 mM*min and 353.25 uM*min. The MAE of the PBPK-WBM ethanol prediction was 0.4096 (compared to Umulis' MAE of 0.291) and the MAE of the acetaldehyde prediction was 2.1230 (compared to Umulis'

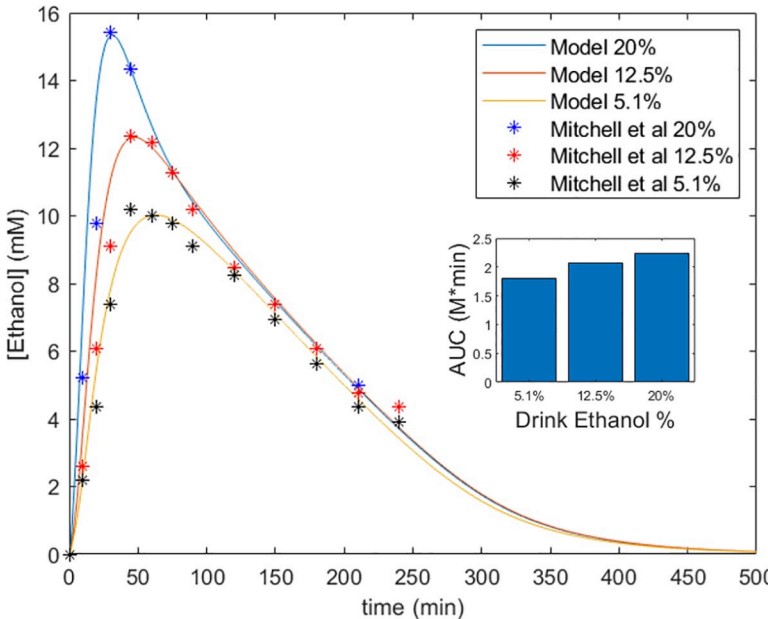

**Fig 3. Effect of ingested ethanol concentration on absorption kinetics.** Our combined model fitted to data from Mitchell et al [47]. We varied absorption ($k_{Stom}$, $k_{SI}$) and transport ($k_{StomSI}$) parameters in the PBPK model with a $V_{max}$ of 1.5mM/min to fit the data for beverages with concentrations of 20%, 12.5%, and 5.1% by a group of men with an average age of 37.8, mass of 82.66 kg, height of 177.1 cm and body fat of 20% drinking 0.5g ethanol/kg. The inset Fig shows the $AUC_{0-\infty}$ for the different ethanol percentages based on the model prediction.

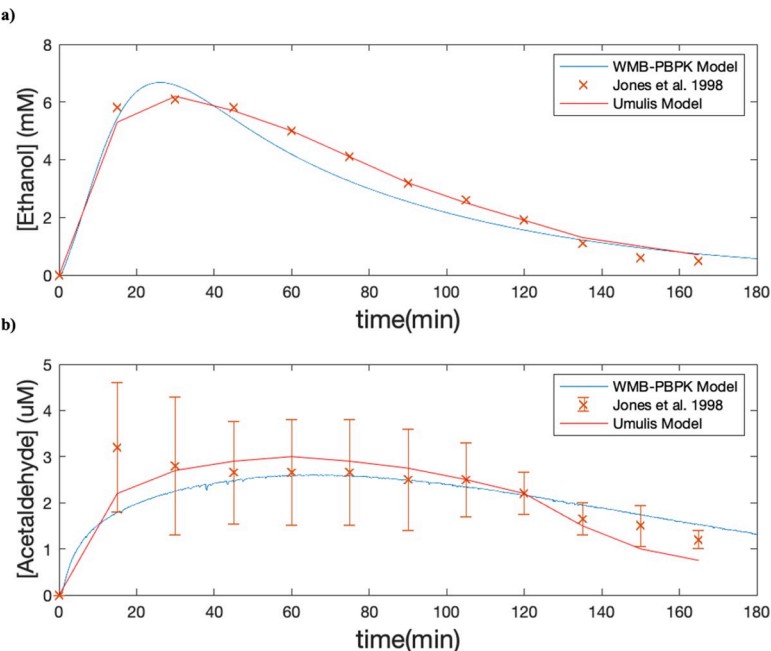

**Fig 4. Comparing the PBPK-WBM model with the data from Umulis et al [11] and Jones et al [48]. (a)** Our gut-absorption fitted model was modified with the Michaelis-Menten kinetics and physiological parameters to match those of a previous developed ethanol metabolism model by Umulis et al. The simulation was performed on a group of men with an average age of 25.6, mass of 74.5 kg, height of 180 cm and body fat of 20% drinking 0.25g ethanol/kg. Simulations were performed with a tolerance of 0.1%. For more information about the effect of model tolerance on prediction, please see Supplemental Information SI 6. **(b)** Predictions made with the fitted model for acetaldehyde concentrations in a simulated male with the same average characteristics.

MAE of 0.1255). Although the fit for the PBPK-WBM model is not as accurate as the Umulis Model, it needs to be taken into consideration that the model used the same Michaelis-Menten parameters as Umulis, while also having additional ethanol metabolism reactions which may lead to discrepancies. A more in-depth analysis is provided in the discussion section.

## Impact of enzyme expression on ethanol metabolism

Given that alcohol dehydrogenase and liver microsomal enzyme oxidation systems are inducible, chronic alcoholics generally have higher levels of enzyme expression, which can lead to decrease the exposure to both ethanol and acetaldehyde [3]. Interestingly, liver cirrhosis is also associated with tissue degeneration and decreased functional mass which would have the opposite effect. As the PBPK-WBM comprehensively captures liver metabolism, we could simulate the effect of the different levels of enzyme expression on elimination by changing the upper bounds of both the liver alcohol metabolism reaction and acetaldehyde reactions. The simulations showed that when liver ethanol elimination was decreased, other processes would carry more flux to compensate. However, the total time to eliminate both ethanol and acetaldehyde increased with high levels of enzyme expression (Fig 5B and 5C). A more in depth analysis of the model's ability to compute the trade-off between enzyme expression and liver mass is provided in the discussion.

## Impact of ALDH isoform on acetaldehyde exposure

Acetaldehyde is primarily metabolized by ALDH2 into acetate, and Chen et al found significant differences in the activity of various ALDH2 isoforms compared to the wildtype (WT)

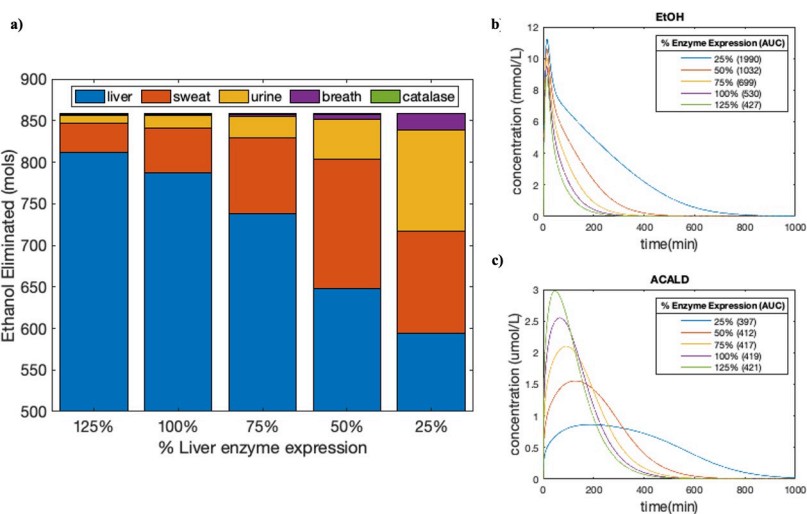

**Fig 5. Effect of enzyme expression on ethanol metabolism. (a)** Total ethanol elimination through various pathways as a result of changing liver enzyme expression. The simulation was performed with the PBPK-WBM model fitted to both Umulis et al. and Mitchell et al. on a group of men with an average age of 25.6, mass of 74.5 kg, height of 180 cm and body fat of 20% drinking 0.25g ethanol/kg for a total of 1000 minutes. See S1 and S2 for tabulated values. **(b, c)** Time-concentration curves of ethanol and acetaldehyde metabolism made with the fitted PBPK-WBM model. The AUC values in the legend have units of mMol*min and uMol*min, respectively.

[49]. More information about the isoforms and their activity can be found in Table 4. Given that the rate of acetaldehyde elimination via ALDH2 is highly dependent on individual genetics, we performed simulations to characterize how the different isoforms could lead to differential exposure to acetaldehyde (Fig 6A).

## Impact of disulfiram on acetaldehyde exposure

Disulfiram is a drug commonly administered to treat chronic alcoholism through inducing the symptoms associated with acetaldehyde build up [50]. To gauge the effect of this treatment on acetaldehyde exposure levels, we used in vitro data presented by Kitson et al (see S3 Text) to identify a correlation between disulfiram concentration and ALDH function and to gauge the effects of disulfiram on acetaldehyde exposure [50]. We used a clinically relevant constant blood disulfiram level of 2-8mg/L to characterize its effect on acetaldehyde elimination [51]. The results show that as the concentration of disulfiram increased, the exposure to acetaldehyde (AUC) and the time for elimination also increased.

## Multi-dosing regimen

Given that ethanol is often consumed in social settings and not in one large dose instantly, we investigated the effects of taking multiple doses of ethanol within a period. We simulated a

**Table 4. Effect of ALDH2 mutations on enzyme activity. In vitro values taken from [48].**

| ALDH2 isoform | Mutation | Allelic frequency | Major ethnicity | % of WT activity |
|---|---|---|---|---|
| ALDH2.1 | WT | | | 100 |
| ALDH2.2 | E504K | 26.6% | East Asia | 1.5 |
| ALDH2.3 | I41V | 0.6% | African | 60 |
| ALDH2.4 | P92T | 2.5% | Latino | 32.5 |
| ALDH2.5 | T244M | 0.4% | South Asian | 36 |
| ALDH2.6 | V304M | 2.7% | Latino | 12.5 |
| ALDH2.7 | R338W | 1.2% | Finnish | 23 |

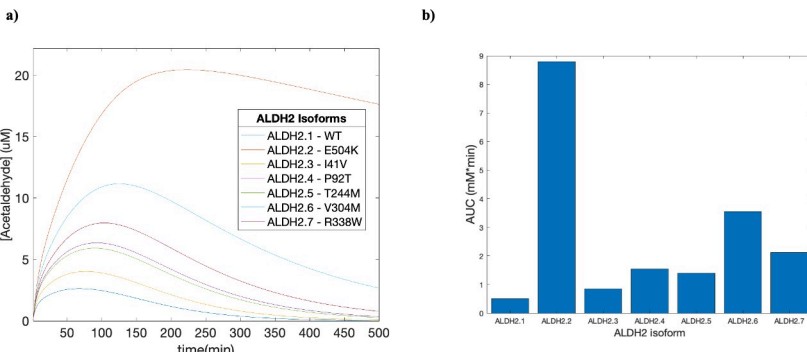

**Fig 6. Effect of ALDH2 mutations on acetaldehyde exposure. (a)** Time-concentration plots for the effect of various ALDH2 mutations on acetaldehyde concentrations were created by changing the enzyme efficiency. The simulation was performed with the fitted PBPK-WBM model on a group of men with an average age of 25.6, mass of 74.5 kg, height of 180 cm and body fat of 20% drinking 0.25g ethanol/kg for a total of 500 minutes. **(b)** Predictions made with the fitted PBPK-WBM model for the total systemic exposure ($AUC_{0-500}$) by various genotypes.

dose every 60 minutes (Fig 8A) and tracked the concentration of both ethanol and acetaldehyde in various tissue compartments (see S2 Fig). Once again, this simulation revealed that exposure to ethanol increased with increasing ethanol concentration of the drink (Fig 8C). Furthermore, because acetaldehyde production was based on ethanol elimination, there was no decrease in acetaldehyde concentration in the 60-minute dosing regimen until the individual stopped drinking and the ethanol concentrations began to decrease (Fig 8B).

## Discussion

### Predicting the effect of drink concentrations

To add in the absorption kinetics in our new gut component, we initially fitted the PBPK model with data from Mitchell et al to gauge the effect of ethanol concentration on absorption. From Fig 3 we can see that the model predictions were able to accurately capture the movement of ethanol through the body with low absolute error. The similarity in $AUC_{0-\infty}$ also shows that the model was consistent with clinical data in representing the overall systemic exposure to ethanol.

This simulation showed that as the concentration of ethanol increased, speed of absorption by the stomach and small intestine increased, which led to a higher maximum concentration faster (Fig 3). This different in absorption rate and systemic distribution was shown by the changes in the shape of the curve near the maximum concentration. After the ethanol was absorbed and distributed, the time-concentration curves behaved similarly regardless of concentration because ADH was operating at $V_{max}$. As the concentration of ethanol fell and the ADH became subsaturated, the Michaelis-Menten kinetics became prominent as indicated by curvilinear elimination below concentrations of 4 mM (Fig 3). Furthermore, drinks with higher ethanol concentrations led to a higher exposure to ethanol as indicated by the increased AUC (Fig 3B).

While the results do not show predictions for higher drink ethanol concentrations, the model inputs could be adjusted to deliver a reasonable estimate. However, it is worth noting that since the correlations were derived from drink ethanol concentrations between 5.1%-20%, extrapolated results must be ultimately validated with clinical data. Nevertheless, the results of this simulation show that our model will be able to accurately capture the ADME of ethanol for a reasonable range of ethanol concentrations.

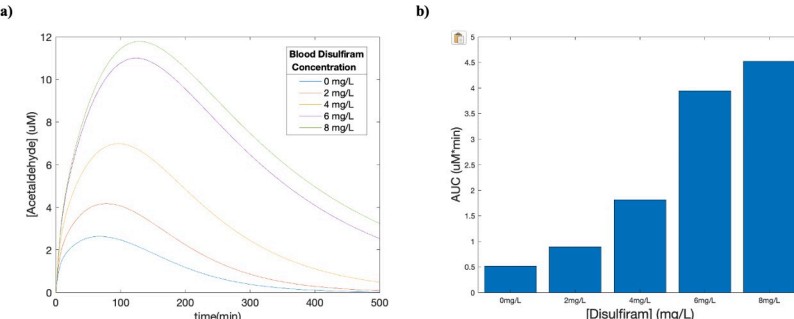

**Fig 7. Effect of blood disulfiram concentrations on acetaldehyde exposure. (a)** Time-concentration plots of the effect of various blood disulfiram levels on acetaldehyde concentrations were created by using correlations generated in vitro from Kitson et al [50]. The simulation was performed with the fitted PBPK-WBM model on a group of men with an average age of 25.6, mass of 74.5 kg, height of 180 cm and body fat of 20% drinking 0.25g ethanol/kg for a total of 500 minutes. **(b)** Predictions made with the fitted PBPK-WBM model for the total systemic exposure to acetaldehyde.

## Predicting acetaldehyde exposure

After ensuring that the gut-absorption component of the model was validated, we sought to validate the acetaldehyde predictions as well. Jones et al. were one of the first groups provide time-concentration data on both ethanol and acetaldehyde, and a previous PK model was built by Umulis et al to capture the ADME of acetaldehyde as well.

One interesting phenomenon in the Jones et al. data was that the peak of acetaldehyde preceded that of ethanol (Fig 4B). Given that acetaldehyde is a metabolite of ethanol, this seems unreasonable. In the original experiment, measurements of both ethanol and acetaldehyde concentrations were taken by end-expired breath gas chromatography analysis [48], and prior to the intake of ethanol the concentration of acetaldehyde was measured to be below the limits of detection, which rules out the possibility of prior ethanol consumption. One possible explanation for this in the Jones et al. paper was that there is the existence of upper-airway microbes that produce acetaldehyde. However, given that the number of subjects in the experiment was only 10 and the error for the early data point was to 41% of the actual data, there is the possibility of a skewed data as well.

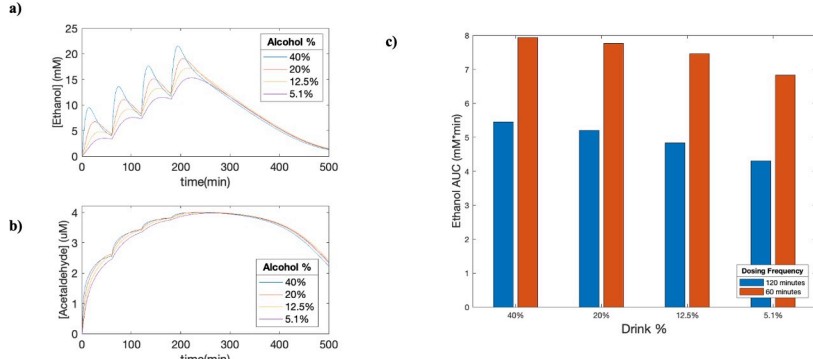

**Fig 8. Effect of multi-dosing on ethanol metabolism (a, b)** Time-concentration plots of various alcoholic drinks on ethanol and acetaldehyde concentrations were created with a dosing frequency of 60 minutes. The simulation was performed with the fitted PBPK-WBM model on a group of men with an average age of 25.6, mass of 74.5 kg, height of 180 cm and body fat of 20% drinking 0.25g ethanol/kg for a total of 500 minutes. **(c)** Predictions made with the fitted PBPK-WBM model for the total systemic exposure to ethanol based on drink concentration and dosing frequency.

Umulis et al. had difficulties characterizing the initial acetaldehyde concentrations, as reflected by the inability of the model to reflect this initial spike in acetaldehyde concentrations (MAE = 0.1255). Given that we used the same Michaelis-Menten kinetic parameters, it was expected that our model would have a similar poor fit. Indeed, the MAE for our model was 2.1230, with the main cause relating to the problematic early data point. The poor fit could indicate that there are other processes in the metabolism of acetaldehyde that have not been explored and taken into consideration in the model as well, although given the comprehensiveness of the WBM this seems unlikely. Alternatively, the Umulis model exclusively focuses on the ADH elimination of ethanol, whereas the integrated PBPK-WBM incorporates the effect of side processes including sweat, urine, breath, and catalase, which will contribute to differences in prediction. Since more ethanol is excreted through non-metabolic pathways, this can explain why our initial predictions were lower than that of Umulis'. In addition, while the Umulis model does a better job of predicting the initial time point, the present model captures the elimination and metabolism in the later time points more accurately.

Limitations of the integrated PBPK-WBM simulation includes utilizing the previously-established Michaelis-Menten parameters which were fitted to data, and a possible solution would be to re-fit the Michaelis-Menten parameters of the PBPK-WBM model to the raw data from Jones et al. In addition, future studies with a larger sample size can be used to update the predictions and draw further conclusions about the metabolism of acetaldehyde, as it is still poorly understood. Furthermore, due to the iterative nature of the PBPK-WBM model, the smoothness of both the acetaldehyde and ethanol curves generated by our model relied heavily on the simulation tolerance, which was set to 0.1%, resulting in 1347 FBA iterations for the 1800 time steps. A smoother curve could be achieved with more iterations, demonstrating the trade-off between accuracy and computational intensity (S3 Fig)

Overall, the results from this simulation show that despite adding additional reactions to comprehensively capture the possible routes of ethanol elimination, the final results do not deviate much from a previously developed model, thus successfully expanding the model for further analysis of specific pathways regulated by gene expression and drugs.

## Impact of enzyme expression on ethanol metabolism

The results of this simulation are interesting for better understanding the relationship between enzyme expression and alcohol usage. As previously stated, chronic alcoholism leads to a higher level of metabolizing enzyme expression, yet also leads to liver cirrhosis and decreased liver mass (hence lower metabolism). In this model, this interesting trade off can be explored since liver mass was calculated based on the individual's physiological characteristics (and can be modified with correlations to account for duration of chronic alcoholism) and expected gene expression. Data on the rate of liver degeneration is not readily available, thus we simulated only the expected effect of enzyme activity on the pathways involved in ethanol metabolism.

From these simulations we see that for a normal level of liver enzyme expression, the mols of ethanol eliminated by the liver accounted for approximately 91% of total ethanol metabolism. As the liver enzyme expression decreases, other processes in the body that eliminate alcohol carried more flux as compensation, shown by the increasing amount of ethanol eliminated by sweat, urine, and breath (Fig 5). Catalase remained a negligible source of ethanol elimination. This trend is likely explained by the fact that at regular enzyme expression levels, sweat and urine elimination would account for at least 3% of the total ethanol metabolism, as set as lower bound in the model (Table 3). However, as the liver deteriorated, the sweat and urine fluxes would increase until they reached the pre-defined upper bound of 10% of total ethanol metabolism rate.

Interestingly, simulations with lower levels of enzyme expression had higher levels of ethanol exposure and slightly lower levels of acetaldehyde exposure as shown by the AUC results in Fig 5B and 5C. This is consistent with clinical observations made by Wicht et al [14], in that a lower ethanol metabolism rate from the liver (and more excretion via breath, urine, and sweat) leads to a lower level of acetaldehyde exposure.

Through this simulation, we demonstrate the capability of the PBPK-WBM model to predict the metabolism of ethanol through non-traditional pathways without needing the specific rates of excretion. The traditional bottom-up development of PBPK models require a large amount of data for each process, yet the WBM can fill the gaps in data that is not readily acquired or cannot be clinically measured. With this novel method of approaching PBPK modelling, more comprehensive models can be made without the need to isolate specific kinetic parameters related to the systems of interest, allowing for a more efficient development of predictive models. Once the specific parameters are eventually measured, the models can be further refined to improve accuracy.

## Impact of ALDH isoform on acetaldehyde exposure

Given that acetaldehyde is a carcinogenic compound, it is important to evaluate the relationship between exposure risk and individuals' genotypes. In the ALDH2.2 isoform, which has been primarily found in the East Asian population, our model predicted a 17-fold increase in exposure in the first 500 minutes when compared to the wildtype (Fig 6B). In other isoforms, we also saw a similar trend that predicted an increase in AUC with decrease in ALDH2 activity. The results of this model agree with current literature, which states that individuals with the ALDH2.2 isoform have a higher exposure to acetaldehyde and additional measures must be taken to minimize potential harmful effects [10].

Limitations of this simulation include that the ALDH2 isoform activities were based on in vitro data, and that the AUC calculated in Fig 6B were only based on the first 500 minutes of data. $AUC_{0-500}$ was used instead of $AUC_{0-\infty}$ because the ALDH2.2 genotype takes over 2000 minutes to reach baseline acetaldehyde levels, which would lead to much larger differences in AUC. Future studies could specifically measure the differences in ALDH activity in clinical studies instead, and a more accurate portrayal of the exposure to acetaldehyde could be generated. Regardless, this simulation shows the advancement of the model to account for the various genotypes in metabolic calculations and improve the personalization of PBPK models.

## Impact of disulfiram on acetaldehyde exposure

Individuals using disulfiram (Antabuse) to treat alcohol dependency can experience higher exposure to acetaldehyde, similar to individuals with the ALDH2.2 isoform. Specifically, the symptoms associated with an increase in blood acetaldehyde concentrations are exactly what drives the individual towards abstinence. In our simulation, we show that as the blood disulfiram concentration increased, the accumulation of acetaldehyde lasted longer and ultimately led to more exposure as shown by the increases in $AUC_{0-\infty}$ (Fig 7B).

A limitation of this simulation includes that the disulfiram concentration was assumed to be at constant steady state, where in reality it would peak after each administration and decrease afterwards. Future iterations of this model can take the elimination kinetics of disulfiram into account and provide a more representative prediction of the exposure to acetaldehyde. Nevertheless, this simulation shows that for individuals undergoing disulfiram treatment for reducing alcohol dependency, they can be subjecting themselves to high levels of acetaldehyde, necessitating a need for future monitoring of long-term carcinogenic effects.

### Multi-dosing regimen

The purpose of the multi-dosing regimen was to modify the model towards more realistic drinking situations. The results from these simulations demonstrate that even when limiting drinks to once every 60-minutes to avoid high ethanol concentrations, people could potentially be exposed to higher levels of accumulated acetaldehyde than expected. As we see in Fig 8B, even though blood ethanol concentrations decrease between drinks, the acetaldehyde continues to accumulate until there is no longer ethanol intake. In this model we included an additional 40% ethanol concentration to simulate stronger drinks that are often taken as shots. The conclusions from the 40% simulation is limited by the fact that it is an extrapolation from the absorption kinetics from Mitchell et al. of 5.1%-20% ethanol concentration [47].

Due to the ability of the PBPK-WBM model to investigate the concentration of metabolites in other tissue compartments as well, the exposure to both ethanol and acetaldehyde in other organs can also be examined for future studies (S2 Fig). It could be particularly interesting to characterize how the exposure to acetaldehyde in various compartments lead to higher risks of developing specific cancers in the future.

### Model limitations

The main goal of this study was to demonstrate a method of PBPK modelling that incorporated a large-scale, organ-resolved WBM to navigate around the challenges associated with acquiring kinetic data. While this model is fitted to clinical data and used to understand the effects of ethanol and acetaldehyde as well as factors affecting their metabolism in the human body, the value lies in the successful integration of a GEM that allows researchers to personalize kinetic parameters without needing specific measurements. Some limitations of the model simulations include the following:

On the PBPK side, the absorption kinetics for ethanol through the gut was fitted based on clinical data ranging from 5.1%-20% drink ethanol concentration. However, given that stronger alcoholic drinks exist, the same correlations can be used but it must be noted that it is an extrapolation from lesser concentrations. Furthermore, the model draws upon the Michaelis-Menten parameters from a previously published study, but also includes additional reactions which cause slight changes in the predictions. A future step here would be to re-fit the Michaelis-Menten parameters based solely on the PBPK-WBM model.

On the WBM side, Since the male Harvey model has an irreversible alcohol dehydrogenase reaction, there is no reverse reaction accounted for, which may have an impact in populations who are not able to metabolize acetaldehyde efficiently [11]. With a build-up of acetaldehyde, the reverse reaction may be involved in lowering the acetaldehyde exposure. However, in the study by Jones et al [48], it was concluded that the reverse reaction does not make a significant contribution.

For the data and parameters used, the fit for the disulfiram inhibition of ALDH2 was created using data from sheep liver cells, which likely behave differently than human liver cells in vivo. Further clinical studies and data could be used to improve the fitting. In the modelling of genetic variations, all the simulated subjects were homozygous for the ALDH2 isoform. In reality, some individuals are likely to be heterozygous, which may affect the effect enzyme elimination rates. Furthermore, the activities of enzyme isoforms were measured in vitro, which may not translate well to in vivo reaction rates.

### Model significance

Traditional PBPK models require a myriad of pharmacokinetic parameters to properly characterize the ADME of various metabolites. Here, we integrate a GEM that bypasses this challenge

through incorporating the pseudo-steady state assumption and dynamic FBA to comprehensively advise on a collection of side reactions. We demonstrate that the combined PBPK-WBM provides reasonable estimates for the ADME of ethanol and acetaldehyde, and additionally allows for the analysis of enzyme expression and genetic factors, an advance from standard PK models. Furthermore, the model expands on previous work to include insight into the effect of drink concentration on ethanol absorption, impact of drugs, multi-dosing regimen, and the accumulation and exposure in various tissue compartments. The success of this framework extends beyond the ethanol modelling space and this technique can be applied to study systems in which there is little kinetic information known, an advance towards better implementing precision medicine.

## Conclusion

In this study, we combined a PBPK model with a GEM to analyze the effects of ethanol and acetaldehyde metabolism. To do so, we added missing transport and excretion reactions to the WBM to achieve optimal overlap between the PBPK and WBM model. The uptake and absorption rates in the model were varied to fit to clinical data based on alcohol concentration, and predictions were made for the effect of drink concentration, organ damage, genetic variations, disulfiram concentration, and dosing regimen. In our simulations, we found that different drink concentrations led to different shapes in the time-concentration curves, indicating different absorption rates and changes in the overall systemic exposure. Furthermore, we explored the effect of changing liver enzyme expression on the activities of other eliminating processes. Through multi-dosing simulations, we find that even at moderate alcohol consumption rates individuals are exposing themselves to resulted in the accumulation of acetaldehyde, which could be explored in the future to gauge the correlation between acetaldehyde levels and pathogenesis.

PBPK models allow for individual tissue-compartment tracking and are the industry standard for drug modelling. However, PBPK models generally only capture the metabolism of the metabolite(s) of interest and require reliable knowledge of kinetic information. In contrast, WBMs can provide a detailed and comprehensive description of human metabolism at the genome scale. However, due to their underlying steady state assumption, they cannot predict concentration changes over time, a key characteristic of PBPK models. Hence, WBMs are useful for predicting steady state concentrations while PBPK models are useful for predicting the dynamics given a set of parameters.

By combining the benefits of both models, we could gain further insight into how genetic variation in one organ could lead to downstream effects in other tissue. In the absence of human data, the combination of models can lead to a mix of approaches to fill in the gaps necessary to fully characterize a metabolic process. While our PBPK-WBM model focused on ethanol and its metabolites, this approach could be applied to other commonly measured metabolites or drugs, thereby enabling a more comprehensive understanding of how different aspects of human metabolism are interconnected. By expanding this framework, it is possible to envision a future where tests and drugs can be given to an individualized *in silico* metabolomic "twin" to ensure safety and efficacy prior to actual administration, ultimately moving towards precision medicine.

## Supporting information

**S1 Fig. Schematic for the dynamic Flux Balance Analysis process for integrating a WBM with a PBPK model.**
(TIF)

**S2 Fig. Time-Concentration curves for ethanol metabolism with a dose every 120 minutes.** Simulations were performed with 20% ethanol for 25.6 year old men weighing 74.5kg with height = 180cm and 20% body fat drinking 0.25g/kg ethanol. **(a)** Multi-dosing curves for ethanol concentration in various tissue compartments. **(b)** Multi-dosing curves for acetaldehyde concentration in various tissue compartments. **(c)** Area Under the Curve for both ethanol and acetaldehyde in the liver.
(TIF)

**S3 Fig. Effect of tolerance on smoothness of model.** As tolerance decreases, the curve better approximates the normal PBPK model. The predictions at 1% and 10% are at lower values when compared to the PBPK model because the WBM model also predicts for methods of ethanol elimination beyond Alcohol Dehydrogenase.
(TIF)

**S1 Table. List of ethanol-related reactions.**
(DOCX)

**S2 Table. List of acetaldehyde-related reactions.**
(DOCX)

**S1 Text. Organ-specific equations.**
(DOCX)

**S2 Text. Correlation between drink concentration and gut absorption.**
(DOCX)

**S3 Text. Correlation between Disulfiram concentration [uM] and ALDH2 activity.**
(DOCX)

## Author Contributions

**Conceptualization:** Ines Thiele, Radhakrishnan Mahadevan.

**Data curation:** Leo Zhu, Radhakrishnan Mahadevan.

**Formal analysis:** Leo Zhu, Radhakrishnan Mahadevan.

**Funding acquisition:** Ines Thiele, Radhakrishnan Mahadevan.

**Investigation:** Leo Zhu, William Pei.

**Methodology:** Leo Zhu, William Pei, Ines Thiele.

**Resources:** Radhakrishnan Mahadevan.

**Software:** Leo Zhu, William Pei, Ines Thiele, Radhakrishnan Mahadevan.

**Supervision:** Radhakrishnan Mahadevan.

**Validation:** Leo Zhu.

**Visualization:** Leo Zhu, Radhakrishnan Mahadevan.

**Writing – original draft:** Leo Zhu, William Pei.

**Writing – review & editing:** Leo Zhu, Ines Thiele, Radhakrishnan Mahadevan.

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
