## [Decision Letter · Decision Letter 0]

27 Jan 2021

Dear Professor Mahadevan,

Thank you very much for submitting your manuscript "Integration of a physiologically-based pharmacokinetic model with a whole-body, organ-resolved genome-scale model for characterization of ethanol and acetaldehyde metabolism" for consideration at PLOS Computational Biology.

As with all papers reviewed by the journal, your manuscript was reviewed by members of the editorial board and by several independent reviewers. In light of the reviews (below this email), we would like to invite the resubmission of a significantly-revised version that takes into account the reviewers' comments.

In addition to a detailed response to the reviewers' comments the revision has to address the novelty of your investigations vs the Umulis 2005 paper (mentioned by one of the reviewers), and the contribution the novel components provide to a new understanding of the metabolism of ethanol.

We cannot make any decision about publication until we have seen the revised manuscript and your response to the reviewers' comments. Your revised manuscript is also likely to be sent to reviewers for further evaluation.

Sincerely,

James Gallo

Associate Editor

PLOS Computational Biology

Feilim Mac Gabhann

Editor-in-Chief

PLOS Computational Biology

Reviewer's Responses to Questions

**Comments to the Authors:**

Reviewer #1: Thanks for the opportunity to review the article. While the model is generally interesting, there are some major issues with the manuscript. Here are my comments for the authors to consider.

1. Please briefly discuss why an input in the lung is necessary in the model, as shown in Figure 1.

2. Please check the units of Figure 3, including Ethanol concentration and AUC in inset figure.

3. The authors established a correlation between ethanol concentration and kinetic rates. However, there is no observed data beyond this range of 5.1% to 20%. Considering the observed range is relatively narrow, extrapolating to 40 and 100% is dangerous, as there is no data to support the same relation exists beyond the 20% limit. Unless the authors could find data to support such a claim, I would suggest removing the simulations at 40% and 100%, as the simulated results could be misleading to the readers.

4. Section 3.2: while prediction of ethanol concentration from Jones et al 1988 is by-and-large ok, the prediction of early phase acetaldehyde clearly misses the observed data. The observed data indicates an abrupt increase of acetaldehyde that’s not captured by the model. However, the time to peak of acetaldehyde seems to be earlier than its parent compound, alcohol, which is a bit hard to believe. I would urge the authors to find other sources of acetaldehyde data and see if they have a better fit and validation of their model. Capturing AUC should not be used as validation to predict profiles over time. Please also discuss the limitation of predicting these profiles.

5. Please discuss whether hepatic impairment will lead to changes in expression or activity of enzymes involved in alcohol metabolism.

6. I do not think changing enzyme activities numerically could be directly related to a percentage change of liver functions, as suggested in Figure 5. Unlike renal impairment where a direct measurement of eGFR etc could give a good estimation of renal function, there is no such value for liver impairment, as grade of which is usually defined by a composite CP score. Please update Figure 5 where “% liver function” is used to something like “% liver enzyme activity” and the related language in text. Same goes for figure SI.3.

7. Figure 5b: the total eliminated ethanol combining all organs are clearly different for different “% liver function”. For example, more ethanol are cleared in the 100% category than that in 10% category. By a simple rule of mass balance, if the same amount of ethanal are given, then the same amount should be eliminated regardless of their liver function. Please explain why this is the case, and if there are other significant sources of elimination, they should be plotted.

8. Why there is no discussion section?

9. Figure SI 8: unclear organ (third figure, top row).

Reviewer #2: This is a well written exposition of the processes the authors used to refine the PK models for ethanol ADME. I do have a few questions that could benefit from a little more discussion/disclosure in the manuscript.

1. On the compartmental model diagram, why is ethanol being introduced into the lung? I understand that ethanol is ingested and not inhaled.

2. When describing the previously published data sets used to inform and test the model, please reference the n of subjects that went into each, so the reader has a better sense of the rigor of data going into the models.

3. With regards to modeling the genetic variants of ALDH2, it needs to be more clearly disclosed that these % efficiency values come from in vitro models. I still think it is OK to use these, but it reads like (or the reader assumes) the % efficiencies came from human data.

4. In the results discussion, there needs to be more emphasis throughout on how your findings differ or improve upon the existing models. As a modeler who is not familiar with this particular literature, it would seem to me that you are arriving upon the same "answer" as previous models, and therefore, what is the point of making a new model? Perhaps this is not the case, but whether you do or don't improve the predictiveness of any one model or if the utility is that you have a model that can predict multiple outcomes, needs to be made clearer. The utility of predictions of how different biological variables would affect ethanol/acetaldehyde exposure is clear and justified, giving a good jumping off point for someone who wants to test this in humans.

Reviewer #3: Overall this paper presents an updated model from previously published models from 16 years ago. While the data is acquired from the Umulis paper, the Umulis paper itself drew the data from the primarily literature and the Authors should rely on the original source data at a minimum to optimize their models. the paper is not a signficant advance as the primary updates are parametric and an increase in the number of compartments for the PBPK. The link to the genome primarily impacts the rates laws for ALDH2 and therefore this is not a significant advance for the community in PBPK or alcohol metabolism.

**Have all data underlying the figures and results presented in the manuscript been provided?**

Reviewer #1: Yes

Reviewer #2: Yes

Reviewer #3: Yes

PLOS authors have the option to publish the peer review history of their article (what does this mean?). If published, this will include your full peer review and any attached files.

Reviewer #1: No

Reviewer #2: No

Reviewer #3: No
---

## [Decision Letter · Decision Letter 1]

24 May 2021

Dear Professor Mahadevan,

We are pleased to inform you that your manuscript 'Integration of a physiologically-based pharmacokinetic model with a whole-body, organ-resolved genome-scale model for characterization of ethanol and acetaldehyde metabolism' has been provisionally accepted for publication in PLOS Computational Biology.

Best regards,

James Gallo

Associate Editor

PLOS Computational Biology

Feilim Mac Gabhann

Editor-in-Chief

PLOS Computational Biology

Reviewer's Responses to Questions

**Comments to the Authors:**

Reviewer #2: The authors have responded to the reviewers' comments.

**Have the authors made all data and (if applicable) computational code underlying the findings in their manuscript fully available?**

Reviewer #2: Yes

PLOS authors have the option to publish the peer review history of their article (what does this mean?). If published, this will include your full peer review and any attached files.

Reviewer #2: No

---

## [Editor Report · Acceptance letter]

29 Jul 2021

PCOMPBIOL-D-20-02155R1 

Integration of a physiologically-based pharmacokinetic model with a whole-body, organ-resolved genome-scale model for characterization of ethanol and acetaldehyde metabolism

Dear Dr Mahadevan,

I am pleased to inform you that your manuscript has been formally accepted for publication in PLOS Computational Biology. Your manuscript is now with our production department and you will be notified of the publication date in due course.

With kind regards,

Andrea Szabo
